# Serum Low Density Lipoprotein Cholesterol Concentration Is Not Dependent on Cholesterol Synthesis and Absorption in Healthy Humans

**DOI:** 10.3390/nu14245370

**Published:** 2022-12-17

**Authors:** Frans Stellaard, Sabine Baumgartner, Ronald Mensink, Bjorn Winkens, Jogchum Plat, Dieter Lütjohann

**Affiliations:** 1Department of Nutrition and Movement Sciences, NUTRIM (School of Nutrition and Translational Research in Metabolism), Maastricht University Medical Center, 6200 MD Maastricht, The Netherlands; 2Institute of Clinical Chemistry and Clinical Pharmacology, University Hospital Bonn, Venusberg-Campus 1, 53127 Bonn, Germany; 3Department of Methodology and Statistics, CAPHRI (Care and Public Health Research Institute), Maastricht University, 6200 MD Maastricht, The Netherlands

**Keywords:** LDL cholesterol, synthesis, absorption, lathosterol, campesterol

## Abstract

Introduction. Pharmacological reduction of cholesterol (C) synthesis and C absorption lowers serum low-density lipoprotein C (LDL-C) concentrations. We questioned whether high baseline C synthesis or C absorption translates into high serum LDL-C concentrations or if there was no connection. Therefore, we studied the association between serum LDL-C and C synthesis or C absorption in healthy subjects. Methods. Three published data sets of young subjects on different diets (study 1), mildly hypercholesterolemic subjects without cardiovascular disease (study 2) and healthy controls of the Framingham study (study 3) were used. The three study populations varied in sex, age, and weight. C synthesis and C fractional absorption rate (FAR) were measured with fecal sterol balance and stable isotope techniques (studies 1 and 2). Additionally, serum lathosterol and campesterol concentrations corrected for the serum total C concentration (R_lathosterol and R_campesterol) were used as markers for hepatic C synthesis and C FAR, respectively (studies 1–3). Linear regression analysis was applied to evaluate associations between LDL-C, C synthesis, and C absorption. Results. Seventy-three, 37, and 175 subjects were included in studies 1, 2, and 3, respectively. No statistically significant associations were found between LDL-C and the measured C synthesis and C FAR, nor for R_lathosterol and R_campesterol in any of the study groups. This lack of associations was confirmed by comparing the male subjects of studies 1 and 2. Study 1 subjects had a 50% lower serum LDL-C than the study 2 subjects (*p* < 0.01), but not a lower C synthesis, C FAR, R-lathosterol, or R_campesterol. Conclusions. Under physiological conditions, C synthesis and C FAR are not major determinants of circulating serum LDL-C concentrations in healthy subjects. The results need to be confirmed in large-scale studies in healthy subjects and patients at risk for cardiovascular disease.

## 1. Introduction

Patients with elevated total serum cholesterol (TC) and in particular low-density lipoprotein C (LDL-C) in combination with additional risk factors for cardiovascular disease (CVD), such as obesity, diabetes mellitus type 2 (DM2), hypertension (HT), and family history with cardiovascular events, are treated by cholesterol-lowering therapies. First, patients need to achieve sufficient reductions in serum LDL-C concentrations and patients with high-risk scores require even more extensive LDL-C lowering to reach LDL-C levels below 1.5 mmol/L or 55 mg/dL [1,2]. LDL-C lowering requires upregulation of the hepatic LDL-receptor, which can be established by a reduction of the hepatic free C pool. The first treatment of choice is a statin treatment, which aims to reduce cholesterol synthesis in the liver by blocking 3-hydroxy-3-methylglutaryl coenzyme A (HMG-CoA) reductase activity in the mevalonate pathway. In the case of insufficient serum LDL-C reduction, even after a dosage increase and/or change in statin, it is generally concluded that the patient’s C synthesis is too low to react sufficiently to statin treatment. As an alternative, ezetimibe treatment may be started to inhibit C absorption, as a high C absorption rate is a second potential cause of elevated serum C. Ezetimibe inhibits the Niemann-Pick C1-Like 1 (NPC1L1) protein that transports sterols into the intestinal cell and reduces the absorption of dietary and biliary C and therewith the hepatic C influx [1]. To maximize the C-lowering effect, a statin and ezetimibe may be combined [3]. It is generally known that C-lowering drugs, and in particular statins, may have significant side effects. Increasing the dose using single drug treatment enhances the side effects. During combination treatment, low dosages of both drugs can be used. Twenty mg of statin is combined with 10 mg of ezetimibe daily. A low-dose combination treatment is more effective than a high-dose single treatment of statin or ezetimibe acid [4,5]. In the case of statin intolerance, statins can be replaced by bempedoic acid [6]. Patients with familial hypercholesterolemia have high serum LDL-C levels, i.e., >5 mmol/L or >190 mg/dL [7]. This may be caused by mutations encoding the LDL-receptor leading to very low receptor activity [8,9] or by hyperactivity or a higher number of proprotein convertase subtilisin/kexin type 9 (PCSK9) [10], which catabolizes the LDL-receptor [11]. In this case, statin and ezetimibe treatments are not sufficient. PCSK9 inhibition is then added [12,13,14].

An interesting observation was made in the Framingham study as described by Matthan et al. [15]. Cases with defined CVD and not taking lipid-lowering drugs were compared with control subjects. The serum LDL-C concentrations were normal to mildly increased and identical in both groups. Thus, in this study, the cardiovascular events were not related to elevated LDL-C concentrations. The authors found that the cases had enhanced serum concentrations of campesterol, sitosterol, and cholestanol corrected for the TC concentration (R_campesterol, R_sitosterol, R_cholestanol), which represent surrogate markers for C absorption [15]. The serum lathosterol concentration corrected for the TC concentration (R_lathosterol), as a surrogate marker for C synthesis was found to be reduced. These results [12] suggest that increased C absorption possibly initiates cardiovascular events without increasing serum LDL-C. This leads to the question of whether LDL-C is related to C synthesis and C absorption. Pharmacological treatment demonstrates that inhibition of synthesis and particular hepatic synthesis by statins and inhibition of absorption by ezetimibe reduces serum LDL-C dose-dependently [3]. However, does the relationship also apply under physiological conditions? The result of the Framingham study may also have been affected by the fact that surrogate serum markers for absorption and synthesis were used instead of a direct measurement of whole-body C synthesis and C absorption. The association between absorption markers and directly measured fractional absorption was questioned under physiological conditions [16]. Do the markers reflect the actual function in the physiological situation or only during pharmacological treatment?

It must be realized that hepatic C synthesis and C absorption affect the hepatic C pool. Reduction of this pool by means of statin and ezetimibe treatment initiates serum LDL-C lowering. However, C synthesis and C absorption are not the only fluxes affecting the hepatic C pool. Other C influxes such as HDL-C and effluxes such as very low density lipoprotein C (VLDL-C), biliary C secretion, and bile acid synthesis, play important roles. Therefore, hepatic C homeostasis is a complex multifactorial process, and, as a consequence, the establishment of the serum LDL-C concentration is also complex. This accentuates the question of whether C synthesis and C absorption play dominant roles in the establishment of the serum LDL-C concentration. Therefore, we aimed to determine the associations between serum LDL-C and C synthesis and C absorption. To study the effects of physiological parameters such as age, sex, and BMI on the potential relationships, we restricted our study population to healthy subjects in order to exclude confounding effects introduced by pathological events. Furthermore, we needed to differentiate between the methods of measuring C synthesis and C absorption. At first, we used the classical fecal sterol balance method [17] and the stable isotope technique [3], respectively. These techniques have been applied only in a limited number of small studies. Secondly, surrogate marker techniques [18,19] have been developed that can be used on a larger scale. In this study, we validated the marker technology for association studies.

## 2. Materials and Methods

Data from three earlier published studies were re-evaluated for these new research questions. In the first study, serum lipids as well as C absorption and C synthesis were measured with original stable isotope tracer techniques and side-by-side with surrogate marker concentrations in young omnivores, lacto-ovo vegetarians, lacto vegetarians, and vegans [20]. The diet compositions have been described in detail in the original publication [17]. Shortly, omnivores ate all kinds of foods. Lacto-ovo vegetarians consumed no meat, fish, or dairy products. Lacto vegetarians did not eat meat, fish, or eggs. Vegans did not consume meat, fish, eggs, dairy products, or honey. Subjects under any medication or intake of dietary supplements fortified with cholesterol-lowering agents such as plant sterol or stanol esters were excluded. In the second study, the same parameters were measured in older mildly hypercholesterolemic subjects under placebo and cholesterol-lowering conditions [21]. Only the data obtained under placebo conditions were included in this study. Both studies were performed at the Institute of Clinical Chemistry and Clinical Pharmacology, University Hospital Bonn, Bonn, Germany. All serum lipids, whole-body cholesterol synthesis, and the fractional cholesterol absorption rate, as well as surrogate plasma markers, were measured in the Laboratory for Special Lipid Analysis at the same institute. Fecal neutral and acidic sterol excretion was measured by gas chromatography-flame ionization detection (GC-FID) [17,18]. C absorption was measured with stable isotope methodology applying GC-mass spectrometry (MS) [17,18,22]. Synthesis and absorption markers in serum were measured with GC-MS [23,24]. The third study was part of the Framingham study [15]. From this study, only the data of controls who did not suffer from diabetes or hypertension were selected. Serum lipids and C synthesis and absorption marker concentrations and their ratios to enzymatically determined cholesterol were analyzed at the Lipid Metabolism Laboratories of the Tufts University School of Medicine, Boston, USA. Synthesis and absorption markers were measured by GC-FID. Studies 1 and 2 allow us to study the associations between LDL-C and synthesis and absorption measured with the original stable isotope tracer methods and with marker technology (lathosterol, campesterol, R_Lathosterol and R_Campesterol), while in study 3, only the synthesis and absorption markers could be used to relate to serum LDL-C.

## 3. Statistical Analysis

Within each study group, the association between LDL-C (dependent variable) and the individual parameters C synthesis, C FAR, and the marker concentrations and ratios (independent variables) were determined by applying linear regression analysis. In study 1, associations were tested in all four diet groups separately: omnivores, lacto-ovo vegetarians, lacto vegetarians, and pure vegans. In studies 1 and 3, associations were also tested in males and females separately. In study 3, the effects of age and BMI on the associations were studied. At first, it was determined whether the slope was significantly different from zero (*p* ≤ 0.05) and whether the slope was positive or negative. Secondly, the R square was determined in order to measure the goodness of fit and to express the potential overall contribution of the tested parameter to the establishment of the height of serum LDL-C concentration. In studies 1 and 2, the mean measured C FAR, synthesis rates, and marker concentrations and ratios were calculated in all subjects. As both studies were performed at the same institute, the data of the males of study 1 were compared with the data of study 2 which was composed of males only. These groups were compared to check whether there was a difference in LDL-C between the groups, which might be accompanied by a difference in C synthesis, C FAR, their marker concentrations and/or ratios to TC. Group comparison was performed using the Mann Whitney U test. *p* ≤ 0.05 was considered significant. GraphPad Prism 8.0.2 was used for all statistical analyses.

## 4. Results

Subject characteristics including serum lipid and lipoprotein concentrations are presented in Table 1. Differences in age, weight, BMI, and serum lipids are observed. Most strikingly, the subjects in study 1 were the youngest, had the lowest weight and BMI, as well as the lowest serum lipid concentrations. Otherwise, subjects in study 3 were the oldest and had the highest BMI and serum triglyceride levels. In Table 2, the results from linear regression analyses are summarized. In a number of study groups, but not all, the lathosterol and/or campesterol concentrations were positively and significantly associated with LDL-C. However, the marker ratios R_lathosterol and R_campesterol, as well as the C FAR and C synthesis rates, were not significantly associated with LDL-C. These lacking associations were not affected by gender, age, and BMI, as subgroup analyses based on these variables showed similar results. The highest R square value was 0.22, found for C FAR. This indicates that neither C FAR nor C synthesis relevantly contributes to serum LDL-C concentrations.

The measured C synthesis and C FAR values, as well as the absolute marker concentrations and ratios to the TC concentration in studies 1 and 2, are included in Table 3 where the males of study 1 are compared with the males of study 2. Significant differences were found for age, weight, BMI, and serum lipids with the exception of high density lipoprotein C (HDL-C). The participants in study 2 had a 50% higher serum LDL-C concentration than the male participants in study 1, and both lathosterol and campesterol concentrations were higher. However, the marker ratios R_lathosterol and R_campesterol, C FAR and C synthesis were not significantly different between both groups. These observations are in agreement with the lack of associations found with linear regression analysis in the individual study groups.

## 5. Discussion

In this study, we tested the associations between serum LDL-C and C synthesis and C absorption under physiological conditions in healthy subjects. In studies 1 and 2, the associations could be studied using measured C synthesis and C absorption, as well as using the marker ratios R_lathosterol and R_campesterol. Applying linear regression, no significant associations could be detected, not for the measured values, nor for the marker ratios. Additionally, a comparison of the male population of study 1 with the subjects of study 2 showed a 50% higher serum LDL-C in study 2. However, the measured C synthesis and C absorption data as well as the marker ratio data were not significantly different in both studies. Studies 1 and 2 were performed in the same hospital and also the same analytical techniques were used. Study 3 was included in order to test the hypothesis in a larger and independent study population. Study 3 was performed in different clinical and laboratory settings. Therefore, we did not perform a group comparison between studies 3 and 1 or 2. It was important to find similar low associations in studies 1 and 2 for the measured synthesis and absorption data and the marker ratios. This process proved the validity of the marker ratios for testing the hypothesis and allowed the interpretation of the regression data for the marker ratios in study 3. Furthermore, in study 3, no significant associations were found between serum LDL-C and the marker ratios R_lathosterol and R_campesterol. The associations were assessed in the whole study group and in males and females, as well as in subjects with different ages and BMIs. The lack of associations was not affected by sex, age, and BMI. The subjects in studies 1 and 2 were all Caucasian. Unfortunately, no data were available for the ethnic background of the subjects of the Framingham study (study 3). This could potentially affect the results. As indicated in Table 2, in some groups, significant positive associations were found between serum LDL-C and the serum lathosterol and/or campesterol concentrations. These associations may be explained by the fact that the markers and C are transported and comparably distributed in the same lipoprotein particles and are similarly affected by the hepatic LDL-receptor activity. Linear regression statistics for the marker ratios need special attention. The marker ratio (Y/X) expresses the ratio between the serum marker concentration (Y) and the serum TC concentration (X). The serum TC concentration is strongly positively associated with the LDL-C concentration as the majority of TC consists of LDL-C. Thus, LDL-C and 1/TC exhibit a negative spurious correlation. As a consequence, a negative relationship between the marker ratio and LDL-C may be expected, even if synthesis and absorption are not associated with LDL-C. However, a lacking association between marker ratio and LDL-C directly proves that C synthesis or C absorption are not determinants of the LDL-C concentration. The R square value obtained with linear regression statistics permits testing whether the X-value adds a relevant contribution to the establishment of the (variation in) Y-value. R square varies from 0 (no fit between the regression line and individual points) to 1 (complete fit). As shown in Table 2, the R square values of all parameters expressing C absorption and C synthesis were low and indicate that FAR and synthesis are not relevant determinants of serum LDL-C under physiological conditions. The lathosterol and campesterol concentrations scored higher R squares, but the highest value of 0.31 for lathosterol is still too low to ascribe a relevant contribution to the size of the serum LDL-C concentration. The relationship between synthesis or FAR with LDL-C has been frequently documented under treatment with a C synthesis or C FAR reducing agent [21,25]. Under physiological conditions and particularly in healthy humans, relationships have not been studied systematically. Surprisingly, our study appears the first to specifically address this important question. Miettinen et al. studied the plant sterols and cholesterol precursors as markers for C FAR and C synthesis in volunteers of a randomly selected Finnish male population [26]. Applying their ratios to total C, no associations were found between the ratios and LDL-C. In another study, Miettinen et al. related the serum cholestanol/cholesterol ratio as a marker for C absorption to lipoprotein C concentrations in 50-year-old-men and found no correlation with LDL-C [27]. Silbernagel et al. described the relationships between C absorption markers and the ATP-binding cassette sub-family G member 5 and G member 8 (ABCG5/G8) alleles in the Ludwigshafen Risk and Cardiovascular health study (LURIC) and the Young Finns Study (YFS) cohorts [28]. In both cohorts, strong associations between frequencies of alleles and the serum marker ratios were found (*p* < 0.0001). However, associations with serum LDL-C were much weaker, being *p* = 0.02 for LURIC and 0.28 for YFS. This indicates a missing link between absorption and serum LDL-C concentrations. Kesäniemi et al. studied the effect of C FAR on serum C in the Finnish population [29] and found a high LDL-C to be associated with a high C FAR and a low C synthesis. C FAR was measured with stable isotope technology. These results were confirmed by Miettinen et al. [26] in 50-year-old-men. The results of the last two studies appear to contradict our results as well as those of the other Finnish studies by the same research group. In vegetarians, reduced cholesterol intake results in only small changes in C FAR, enhanced C synthesis, and reduction of LDL-C [20,30]. This identifies the intestinal C flux as an additional parameter. We considered it important to test the associations using measured whole-body synthesis and FAR values first and in a second line with non-cholesterol sterols as markers for synthesis and FAR. The validity of markers may be dependent on many experimental conditions, as shown by Quintao [31].

Whole body C synthesis and C absorption are the two influxes into the endogenous C pool. They are considered to be determinants of serum LDL-C concentration. The serum TC concentration is the result of the C homeostasis process [32]. Absorbed C enters the liver via chylomicron remnant particles formed from chylomicrons secreted by the intestinal cells. Also, C extracted by the liver from VLDL remnants, LDL and HDL, enters the liver. Thirdly, the liver is the main contributor to whole-body C synthesis. The hepatic C effluxes are biliary C secretion and bile acid synthesis. The liver must coordinate the influxes and effluxes to keep the whole-body C pool constant. C synthesis is controlled in order to balance variation in C absorption. This statement has been proven by pharmacological data showing increased synthesis under ezetimibe treatment [19] and under extreme situations under physiological conditions. Excessive high dietary C intake and absorption may be compensated by reduced C absorption and C synthesis, as well as an increased bile acid synthesis, as shown elegantly in a man consuming 25 eggs every day [33] not suffering from hypercholesterolemia. A very low dietary C intake as in pure vegans leads to only a moderately lower serum LDL-C caused by an increased compensatory synthesis [20]. The mechanism by which the regulation takes place has still not been clarified and may not function efficiently in all subjects. Enhanced C absorption results in an increased hepatic C pool, which may be compensated by decreased hepatic C synthesis. However, also hepatic LDL-C uptake may be decreased, and VLDL-C secretion, biliary C secretion, and bile acid synthesis increased. In this extreme situation, hepatic C synthesis may remain unaltered. The sequence and extent of responsive events are unknown (for a general review of the hepatic C homeostasis, see reference [32]). The serum LDL-C concentration is established by a number of C fluxes, such as hepatic VLDL secretion, conversion of VLDL to LDL, exchange of C ester with HDL, and hepatic and extrahepatic uptake of LDL by the LDL-receptor. The hepatic C pool may be the regulator in hepatic LDL uptake [23]. Statin treatment and ezetimibe treatment share the fact that the hepatic C influx is reduced and therewith the hepatic C pool. Under statin treatment, two compensating effects to restore this pool is to lower hepatic VLDL-C secretion and enhance C extraction from the blood by upregulation of the LDL-receptor activity. Ezetimibe treatment reduces the hepatic influx of chylomicron-derived absorbed C. This effect is compensated by enhanced LDL uptake and enhanced hepatic C synthesis [23]. The original publication of study 2 [18] and the follow-up studies show that the effects of 20 mg simvastatin daily and 10 mg ezetimibe [23,34] daily have different effects on whole-body C synthesis and on R_lathosterol. R_lathosterol has been defined as a marker of hepatic C synthesis, as it reflects the expected effects of simvastatin and ezetimibe on hepatic C synthesis more closely. This also predicts that hepatic and extrahepatic synthesis are affected differently. Extrahepatic C synthesis appears increased under both simvastatin and ezetimibe treatment. It is unclear whether this hepatic balancing plays any role under physiological conditions in healthy subjects. In our study, we did not observe conflicting data for whole-body C synthesis and R_lathosterol in the relationship with serum LDL-C. The Framingham data indicated enhanced C absorption and reduced hepatic C synthesis in the case group [15]. This combination of data might explain the normal serum LDL-C concentration. However, on average, the absorption markers were only 6 to 16 % higher in the cases compared with controls, and R_lathosterol was only 15% lower. It remains unclear whether such small changes in C absorption can introduce cardiovascular events. The cases group had a statistically significantly higher number of patients with diabetes and beta-blocker users. These uninvestigated factors may have played a potentially causal role in the increased development of cardiovascular disease. Diabetes is a known risk factor for the development of atherosclerosis. Beta-blockers have been developed to treat abnormal heart rhythms and are also effective in the treatment of high blood pressure, which is also a known risk factor for cardiovascular disease. The direct effects of beta-blockers on cardiovascular disease are not clearly defined [35].

Our study aimed to test the hypothesis that synthesis and absorption are determinants of LDL-C concentration under physiological conditions. Our results do not support this notion. A high C synthesis or a high C absorption were not linked to a high serum LDL-C concentration. The lacking associations may be explained by the inversed relationship between synthesis and absorption. Our findings may be restricted to the healthy condition of our study populations. A drawback of our study is also the small sample size. Repetition of our calculations in a much larger group of healthy subjects is necessary. However, the associations must also be tested in less healthy subjects with an increased risk to develop cardiovascular diseases in whom the balance between synthesis and absorption is potentially disturbed.

It may be expected that patients with a high C synthesis are more able to reduce their synthesis rate under statin therapy, while patients with high C absorption may more strongly reduce their FAR under ezetimibe treatment. Whether these enhanced reductions directly lead to enhanced serum LDL-C reductions cannot be predicted. Using literature data from studies involving statin and ezetimibe treatment, Descamp et al. could not prove this concept [36]. As indicated by the Framingham data, cardiovascular events may be caused by a factor independent of the serum LDL-C level [15]. Thus, serum LDL-C-lowering and the prevention of CVD development may be two different goals that can be reached in parallel in many, but not all patients.

## 6. Conclusions

This study indicates that C synthesis and C absorption are not major determinants of the serum LDL-C concentration in healthy subjects, which means that either a high C synthesis or high C absorption does not automatically translate into a high serum LDL-C concentration. The results need to be confirmed in large-scale studies in healthy subjects and patients at risk for cardiovascular disease.

## Figures and Tables

**Table 1 nutrients-14-05370-t001:** Subject characteristics of the three study populations. The data are expressed as number of subjects or as mean ± standard deviation. TC = total cholesterol, LDL-C =low density lipoprotein cholesterol, HDL-C =high density lipoprotein C, TG = triglycerides).

	Study 1	Study 2	Study 3
N	73	37	175
Sex	37 F, 36 M	M	50 F, 125 M
Age (years)	25 ± 3	41 ± 8	64 ± 8
Weight (kg)	67 ± 13	84 ± 10	82 ± 16
BMI (kg/m^2^)	22 ± 3	25 ± 2	28 ± 5
TC (mg/dL)	179 ± 28	233 ± 28	204 ± 34
LDL-C (mg/dL)	105 ± 22	157 ± 22	129 ± 31
HDL-C (g/dL)	57 ± 14	53 ± 13	48 ± 15
TG (mg/dL)	89 ± 28	118 ± 43	141 ± 101

**Table 2 nutrients-14-05370-t002:** R_square data for the linear regression evaluating associations between serum LDL-C concentrations and various parameters expressing C synthesis and C absorption.

	Lathosterol	Campesterol	R_Lathosterol	R_Campesterol	Synthesis	FAR
Study 1 Omnivores (*n* = 19)	**0.32 (+)**	0.13	0.04	0.03	0.06	0.01
Study 1 Lacto-ovo vegetarians (*n* = 18)	**0.27 (+)**	0.20	0.01	0.04	0.02	0.01
Study 1 Lacto vegetarians (*n* = 17)	0.03	0.03	0.12	0.09	0.02	0.22
Study 1 Pure vegans (19)	0.09	0.03	0.03	0.04	0.18	0.04
Study 1 Females (37)	0.03	**0.12 (+)**	0.01	0.01	0.06	0.05
Study 1 Males (36)	0.03	0.00	0.03	0.01	0.04	0.03
Study 2 All subjects (*n* = 37)	0.00	**0.20 (+)**	0.04	0.06	0.02	0.10
Study 3 All subjects (*n* = 171)	**0.05 (+)**	**0.13 (+)**	0.00	0.00		
Study 3 Females (*n* = 49)	0.00	0.01	0.07	0.05		
Study 3 Males (*n* = 171)	**0.10 (+)**	**0.19 (+)**	0.00	0.00		
Study 3 Age 30 to 59 years (*n* = 48)	**0.20 (+)**	**0.16 (+)**	0.00	0.02		
Study 3 Age 70 to 82 years (*n* = 53)	0.06	**0.26 (+)**	0.00	0.03		
Study 3 BMI 20 to 25 (*n* = 48)	0.01	0.06	0.01	0.04		
Study 3 BMI 30 to 47 (*n* = 54)	0.04	**0.10 (+)**	0.03	0.00		

Bold numbers indicate that the slope was significantly different from zero. (+) indicates a positive association with LDL-C. R square varies from 0 (no fit between line and points) to 1 (perfect fit). FAR = fractional absorption rate (%). Due to the large number of participants in study 3, we were able to test the effects of age, sex and BMI on the linear regression data.

**Table 3 nutrients-14-05370-t003:** Comparison of subject characteristics, serum lipids, marker concentrations, marker ratios, FAR and synthesis for male subjects of study 1 and 2. Data are expressed as number of subjects or as median and interquartile range (25th and 75th percentile). The statistical difference was tested with the Mann Whitney U Test. TC = total cholesterol, LDL-C =low density lipoprotein cholesterol, HDL-C =high density lipoprotein C, TG = triglycerides), Lath = lathosterol, Camp = campesterol, FAR = fractional absorption rate (%).

	Study 1	Study 1 Males	Study 2	*p*
N	73	36	37	
Sex	37 F, 36 M	36M	37M	
Age (years)	25 (23 to 27)	25 (24 to 26)	41 (37 to 47)	<0.01
Weight (kg)	64 (58 to 74)	73 (65 to 81)	84 (75 to 91)	<0.01
BMI (kg/m^2^)	21 (20 to 23)	22 (20 to 24)	25 (24 to2 7)	<0.01
TC (mg/dL)	180 (159 to 200)	171 (149 to 191)	231 (213 to 252)	<0.01
LDL-C (mg/dL)	100 (89 to 124)	101 (84 to 121)	155 (139 to 174)	<0.01
HDL-C (g/dL)	54 (47 to 67)	50 (43 to 56)	50 (46 to 54)	0.98
TG (mg/dL)	91 (67 to 107)	93 (64 to 114)	109 (88 to 144)	<0.01
Lath (mg/dL)	0.26 (0.18 to 0.35)	0.28 (0.19 to 0.36)	0.37 (0.29 to 0.44)	<0.01
Camp (mg/dL)	0.34 (0.24 to 0.42)	0.34 (0.24 to 0.43)	0.58 (0.33 to 0.70)	<0.01
R_Lath (mg/g)	1.5 (1.1 to 1.9)	1.6 (1.3 to 2.2)	1.7 (1.1 to 2.1)	0.56
R_Camp (mg/g)	1.9 (1.5 to 2.6)	2.2 (1.6 to 2.7)	2.5 (1.4 to 2.9)	0.79
FAR (%)	46 (40 to 57)	47 (41 to 58)	52 (43 to 58)	0.59
Synthesis (mg/d)	917 (752 to 1126)	1049 (832 to 1360)	854 (686 to 1328)	0.14

## Data Availability

No publicly archived datasets have been used. The applied internal data sets may be requested from the authors of this study.

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
