# Peer review of "Serum Low Density Lipoprotein Cholesterol Concentration Is Not Dependent on Cholesterol Synthesis and Absorption in Healthy Humans"

_nutrients, 2022, doi:10.3390/nu14245370_

Round 1

Reviewer 1 Report (New Reviewer)

I congratulate the authors for being very innovative and essential for society research.

here are my suggestion to improve the work:

1) line 72 lathosterol is a surrogate marker, better precise this.

2)In the introduction indicating the significant side effects of the cholesterole-lowering drug is essential to the key point in this issue.

3)line 109 please explain better briefly the type of diet at minimum in macronutrient composition.

4)line 111, please indicate the drugs used here briefly,

5)insert the differences by 3 studies here line 154, e.g. study 3 all patients have value borderline.

6) Table 2 graphically shifts or moves on page 5 the description the table 2 , is more comprehensible.

7) line 270, introduce the concept of people differential in cholesterol respondent and not respondent, there are some reviews and papers.

8)line 298-300 better highlight this issue.

Have a good work, the referee

Author Response

I congratulate the authors for being very innovative and essential for society research.

Answer. We thank the reviewer for the positive response to our manuscript.

here are my suggestion to improve the work:

Comment 1)line 72 lathosterol is a surrogate marker, better precise this.

Answer. The lines 69 to 73 have been adapted as follows: The authors found that the cases had enhanced serum concentrations of campesterol, sitosterol and cholestanol corrected for the TC concentration (R_campesterol, R_sitosterol, R_cholestanol), that represent surrogate markers for C absorption [12]. The serum lathosterol concentration corrected for the TC concentration (R_lathosterol), as a surrogate marker for C synthesis, was found to be reduced.

Comment 2) In the introduction indicating the significant side effects of the cholesterole-lowering drug is essential to the key point in this issue.

Answer. We don’t believe that the indication of the side effects of cholesterol lowering agents improves our manuscript. However, we decided to include the following statement in the introduction: “It is generally known that C lowering drugs, and in particular statins may have significant side effects. Increasing the dose using single drug treatment enhances the side effects. During combination treatment low dosages of both drugs can be used. Twenty mg of statin is combined with 10 mg ezetimibe daily.  A low dose combination treatment is more effective than a high dose single treatment of statin or ezetimibe [4,5]. In case of statin intolerance, Statins can be replaced by bempedoic acid [6]”

Comment 3)line 109 please explain better briefly the type of diet at minimum in macronutrient composition.

Comment 4)line 111, please indicate the drugs used here briefly,

Answers to comments 3 and 4 have been combined. The following statement has been added into the manuscript. “The diet compositions have been described in detail in the original publication [20]. Shortly, omnivores ate all kind of foods. Lacto-ovo vegetarians consumed no meat, fish or dairy products. Lacto vegetarians did not eat meat, fish or eggs. Vegans did not consume meat, fish, eggs, dairy products or honey. Subjects under any medication or intake of dietary supplements fortified with cholesterol lowering agents such as plant sterol or stanol esters were excluded.”

Comment 5) insert the differences by 3 studies here line 154, e.g. study 3 all patients have value borderline.

Answer. The following statement has been added into the manuscript: “ Otherwise, subjects in study 3 were the oldest, had the highest BMI, and highest serum triglyceride levels.”

Comment 6) Table 2 graphically shifts or moves on page 5 the description the table 2 , is more comprehensible.

Answer. Table 2 needs to be formatted. This will be done in collaboration with the layout experts of the journal after the paper has been accepted for publication.

Comment 7) line 270, introduce the concept of people differential in cholesterol respondent and not respondent, there are some reviews and papers.

Answer. In our opinion, from line 259 on we described the concept of the hepatic C homeostasis balancing the various C fluxes including the fluxes of absorbed C and C synthesized in the liver. We stated that “The mechanism by which the regulation takes place has still not been clarified and may not function efficiently in all subjects.” We added the following statement into the manuscript: “Enhanced C absorption results in an increased hepatic C pool, which may be compensated by decreased hepatic C synthesis. However, also hepatic LDL-C uptake may be decreased and VLDL-C secretion, biliary C secretion and bile acid synthesis increased. In this extreme situation hepatic C synthesis may remain unaltered.  The sequence and extent of responsive events is unknown (for a general review about the hepatic C homeostasis see reference [32]).”

Comment 8)line 298-300 better highlight this issue.

Answer. We thank the authors for touching this issue. At first we realize that mentioning diabetes and high glucose concentrations coincide. Therefore we removed the high glucose concentrations. “The cases group had a statistically significant higher number of patients with diabetes and beta-blocker users. These uninvestigated factors may have played a potential causal role in the increased development of cardiovascular disease. “ Thereafter we added the following information: “Diabetes is a known risk factor for development of atherosclerosis. Beta-blockers have been developed to treat abnormal heart rhythms and are also effective in the treatment of high blood pressure, which is also a known risk factor for cardiovascular disease. The direct effects of beta-blockers on cardiovascular disease are not clearly defined [35]”.

Have a good work, the referee

Reviewer 2 Report (Previous Reviewer 3)

The authors have addressed all of my concerns in their revisions.  There are just two very small but important changes that need to be made.

1) Title should read: "Serum low density lipoprotein cholesterol concentration independence of cholesterol synthesis and absorption in healthy humans".  The space between "in" and "dependence" suggests an opposite finding.

2) First line of conclusions in abstract: "Under physiological conditions, C synthesis and C FAR do not appear to be major determinants of circulating serum..."

Otherwise the manuscript is sound.

Author Response

The authors have addressed all of my concerns in their revisions.  There are just two very small but important changes that need to be made.

Answer. We are happy that we were able to convince the reviewer.

Comment 1) Title should read: "Serum low density lipoprotein cholesterol concentration independence of cholesterol synthesis and absorption in healthy humans".  The space between "in" and "dependence" suggests an opposite finding.

Answer. The concerns of earlier reviewers about the small sample size of our study led to the academic editors advice to reduce the strength of statements in the title and conclusions. For that reason we neutralized the title by stating "Serum low density lipoprotein cholesterol concentration in dependence of cholesterol synthesis and absorption in healthy humans". We propose to not change the title.

Comment 2) First line of conclusions in abstract: "Under physiological conditions, C synthesis and C FAR do not appear to be major determinants of circulating serum..."

Answer. Thank you! We changed the sentence as proposed by the reviewer.

Otherwise the manuscript is sound.

This manuscript is a resubmission of an earlier submission. The following is a list of the peer review reports and author responses from that submission.

Round 1

Reviewer 1 Report

This article is not well-written and has many major defects.

1) the research is of limited significance, and there is no good description of the research background in the introduction section. Some statements lack logical relationships. For example, page 2 line 69-72. 

2)This manuscript is only a simple analysis of three small sample studies, and the sample size included in the study is very small, which affects the reliability of the results. The tables in the results section are also not standardized.

3)There are some mistakes of the format in the manuscript. For example, Table 1 and Table 3.

Reviewer 2 Report

The authors seek to show that in 3 previous studies of "mildly hypercholesterolemic" patients that synthesis and absorption of cholesterol are not determinants of serum cholesterol levels.  As this is a re-examination of previous work the novelty is somewhat limited, however the methods are sound and the conclusions are supported by the included data.  

The major item that stands out is a English language issue- "are no determinants" is not grammatically correct.  "are not determinants" or "do not determine" is the correct way to state this.

What is missing is the context around these studies- or rather the studies should be examined better in the context of hypercholesterolemia.  The authors share no explanation for example of why this observation, drawn from studies not designed to analyze these parameters, are conducted in "normal controls" or a population with "mild" hypercholesterolemia (no parameters defined).  Basically, the authors state that in a healthy individual neither consumption of cholesterol (and subsequent absorption) or synthesis are correlated with serum ldl cholesterol levels- but to actually show this one would have to show that inhibition of the synthesis pathway or dietary changes fail to reduce serum ldl levels in the actual patient population.  There needs to be some explanation included of what normal levels are.

Of further concern is the graph showing that the study populations are significantly different.  Im not sure of the end goal of figure 3.  If the study populations used for the analysis are indeed significantly different from each other in almost all metrics, are these the right studies to include in an analysis?  Is it possible that the lack of correlations you use to make your associations are a result of the significantly different populations?  Additionally, as the authors state that study 3 is different enough that they cannot compare it to the other two- yet assume that a lack of associations in this data is important.  If the authors concede that the methodical differences in measurements make the studies incomparable, then why does a lack of association among these variables in an incomparable study thus become important.  If you cannot compare the studies to each other due to method differences then is it acceptable to argue that a lack of certainty (ie significance in correlation) then become comparable to a lack of confidence in the correlations of the other studies.  An explanation of this would clear up a good bit of confusion from this reviewer.

At this time, I just cannot understand the author's take home message.  The data supports the conclusions (as presented) - but is this arguing that statins and absorption blocking drugs do not control serum cholesterol?  Or that they dont work in individuals without clinically diagnosed hypercholesteremia (odd argument), or is it simply in healthy populations there is no way to predict ldl levels using synthesis and absorption metrics (which is admittedly not the most straightforward way to determine LDL cholesterol in the first place when the assay is so well accepted and included in most accepted lipid panels?  

It is this reviewers sincere opinion that I cannot reccommend publication without actually understanding the message that the authors are trying to convey.  I do not see anything that stands out as a major issue in the data itself, but to put it simply without being able to draw a conclusion about what the authors are saying I cannot endorse the article- for no better reason than I do not understand it.  I would request that more information be added to address the above concerns about interpretation.

Reviewer 3 Report

The manuscript of Stellaard et al. describes the analysis of three separate studies examining cholesterol synthesis, uptake and plasma LDL-cholesterol (LDL-C) concentration. Measurement of cholesterol synthesis and uptake were measured indirectly by measurement of fecal sterols and stable isotope methods. These methods are standard in the field and are reliable. The study populations are of young and middle aged, and older moderately healthy individuals (mostly male). The statistics performed were satisfactory. The conclusions are supported by the data. I have a few comments.

1) The references are limited and could be expanded to include many more publications that discuss LDL-C concentration and cholesterol synthesis and uptake.

2) The study populations are a limitation. The relative young age and good health of these populations (Study 1 and 2) need to be stated clearly and multiple times (for example in the title). In these populations, there is no association between LDL-C and cholesterol synthesis and uptake. That is clear. However, I wonder if this association would hold up for an older less healthy population where innate regulatory mechanisms are not functioning well (synthesis is not balanced with uptake).

3) Based on this manuscript, the authors have concluded that LDL-C plasma concentrations do not associate with cholesterol synthesis and uptake. Do LDL-C levels correlate with VLDL secretion rates? What are VLDL secretion rates dependent on? The authors can speak to this in the discussion. Furthermore, it has been shown in other studies that plasma sugar and liver TG levels do associate with higher VLDL secretion rates. I think the authors could have at least added an association calculation between LDL-C and TG (Table 2) and then commented on it further.

3. The authors should reiterate the ethnic origin of the study populations as this also plays a role.

Round 2

Reviewer 1 Report

Although the authors made some changes to the manuscript, I insist on my opinion that this manuscript is only a re-analysis of correlation from small sample data, and the results lack credibility and novelty.

Reviewer 2 Report

I would first like to thank the authors for their well thought out and detailed response to the original review.  All of my original confusion about the work has been clarified.  Unfortunately, my confusion was a result of my own expectations of a more impactful work.  After the dust has settled what is left is a paper that analyzes a small subset of small studies in a "novel" examination, which fails to elicit any excitement whatsoever in this reviewer.  The main take home is that measurements of cholesterol biosynthesis or absorption are unreliable to predict the level of serum cholesterol in a healthy individual.  This message is better suited to a micro-publication, as it simply does not have enough impact (if it was a more exciting finding the lack of substantial original work or data collection  by the authors could be overlooked) to warrant publication as a stand alone article.  This is a story better told as a panel in a larger piece of work.

In conclusion, this reviewer does not find this work compelling enough to overcome the simplistic nature of the study's design or its lack of interest to readers to recommend publication.